# On the Seasonal Dynamics of Phytoplankton Chlorophyll-*a* Concentration in Nearshore and Offshore Waters of Plymouth, in the English Channel: Enlisting the Help of a Surfer

Elliot McCluskey [1], Robert J. W. Brewin [1,2,*], Quinten Vanhellemont [3], Oban Jones [2], Denise Cummings [2], Gavin Tilstone [2], Thomas Jackson [2], Claire Widdicombe [2], E. Malcolm S. Woodward [2], Carolyn Harris [2], Philip J. Bresnahan [4], Tyler Cyronak [5] and Andreas J. Andersson [6]

1  Centre for Geography and Environmental Science, College of Life and Environmental Sciences, University of Exeter, Penryn Campus, Penryn TR10 9FE, Cornwall, UK; em587@exeter.ac.uk
2  Plymouth Marine Laboratory, Plymouth PL1 3DH, Devon, UK; obj@pml.ac.uk (O.J.); dgcummings@hotmail.co.uk (D.C.); ghti@pml.ac.uk (G.T.); thja@pml.ac.uk (T.J.); clst@pml.ac.uk (C.W.); emsw@pml.ac.uk (E.M.S.W.); caha@pml.ac.uk (C.H.)
3  Royal Belgian Institute of Natural Sciences, Operational Directorate Natural Environments, Vautierstraat 29, 1000 Brussels, Belgium; qvanhellemont@naturalsciences.be
4  Department of Earth and Ocean Sciences, University of North Carolina Wilmington, Wilmington, NC 28403-5944, USA; bresnahanp@uncw.edu
5  Department of Marine and Environmental Sciences, Nova Southeastern University, Dania Beach, FL 33004, USA; tcyronak@nova.edu
6  Scripps Institution of Oceanography, University of California, San Diego, CA 92037, USA; aandersson@ucsd.edu
*  Correspondence: r.brewin@exeter.ac.uk; Tel.: +44-1326-255119

**Abstract:** The role of phytoplankton as ocean primary producers and their influence on global biogeochemical cycles makes them arguably the most important living organisms in the sea. Like plants on land, phytoplankton exhibit seasonal cycles that are controlled by physical, chemical, and biological processes. Nearshore coastal waters often contain the highest levels of phytoplankton biomass. Yet, owing to difficulties in sampling this dynamic region, less is known about the seasonality of phytoplankton in the nearshore (e.g., surf zone) compared to offshore coastal, shelf and open ocean waters. Here, we analyse an annual dataset of chlorophyll-*a* concentration—a proxy of phytoplankton biomass—and sea surface temperature (SST) collected by a surfer at Bovisand Beach in Plymouth, UK on a near weekly basis between September 2017 and September 2018. By comparing this dataset with a complementary *in-situ* dataset collected 7 km offshore from the coastline (11 km from Bovisand Beach) at Station L4 of the Western Channel Observatory, and guided by satellite observations of light availability, we investigated differences in phytoplankton seasonal cycles between nearshore and offshore coastal waters. Whereas similarities in phytoplankton biomass were observed in autumn, winter and spring, we observed significant differences between sites during the summer months of July and August. Offshore (Station L4) chlorophyll-*a* concentrations dropped dramatically, whereas chlorophyll-*a* concentrations in the nearshore (Bovsiand Beach) remained high. We found chlorophyll-*a* in the nearshore to be significantly positively correlated with SST and PAR over the seasonal cycle, but no significant correlations were observed at the offshore location. However, offshore correlation coefficients were found to be more consistent with those observed in the nearshore when summer data (June–August 2018) were removed. Analysis of physical (temperature and density) and chemical variables (nutrients) suggest that the offshore site (Station L4) becomes stratified and nutrient limited at the surface during the summer, in contrast to the nearshore. However, we acknowledge that additional experiments are needed to verify this hypothesis. Considering predicted changes in ocean stratification, our findings may help understand how the spatial distribution of phytoplankton phenology within temperate coastal seas could be impacted by climate change. Additionally, this study emphasises the potential for using marine citizen science as a platform for acquiring environmental data in otherwise challenging regions of the ocean, for understanding ecological indicators such as phytoplankton abundance and phenology. We discuss the limitations of our study and future work needed to explore nearshore phytoplankton dynamics.

**Keywords:** phytoplankton; chlorophyll-*a*; phenology; citizen science; coastal; nearshore; surfers

## 1. Introduction

Phytoplankton are regarded as an integral part of the marine ecosystem, owing to their role as the primary producers of the sea and considering their profound influence on Earth's biogeochemical cycles [1]. Despite constituting less than 1% of the Earth's primary producer biomass [2], phytoplankton are responsible for approximately half of global primary production [3,4]; and in the ocean alone, are responsible for 90% of primary production [5]. The disproportionately large role that phytoplankton play in primary production impacts the $CO_2$ concentration and pH of the oceans which together with physical processes, influences air–sea $CO_2$ gas exchange [6]. In the context of a changing planet, the relative importance of phytoplankton and the justification to study them could not be more clear [7].

Phytoplankton growth is controlled by both abiotic and biotic factors [4]. Temperature, light, and the supply of limiting nutrients—principally nitrogen, phosphorous, silicate and iron [8,9]—are thought to be the main abiotic factors controlling phytoplankton growth [2,10,11]. Herbivorous grazing—principally by microzooplankton (20–200 μm in size) that consume 60–70% of global daily phytoplankton production [12,13]—is thought to be a major biotic controlling factor of phytoplankton growth; other factors include disease (e.g., viral lysis [14]). Like plants, phytoplankton exhibit seasonal and interannual cycles driven by physical, chemical, and biological processes [15–23].

The concentration of chlorophyll-*a* (chl-*a*) pigments, the principal photosynthetic pigments in phytoplankton (representing the sum of mono- and divinyl chlorophyll-*a*, chlorophyllide-*a*, and the allomeric and epimeric forms of chlorophyll-*a*), is often used as a proxy of phytoplankton biomass, given it is in all phytoplankton and is relatively easy to measure. Common methods for measuring chl-*a* include filtering seawater, extracting phytoplankton pigments using solvents, and measuring chl-*a* concentration *in vitro* using fluorometry, spectrophotometry or through high-performance liquid chromatography (HPLC); using *in-vivo* fluorometric sensors, or flow-through absorption or attenuation-based spectrophotometric sensors, integrated into ship underway systems, ship vertical profilers and ocean moorings [24–30]. More recently, the use of ocean robotics platforms, such as Biogeochemical-Argo floats and ocean gliders, is expanding data coverage of chl-*a* [31,32]. Due to the optical characteristics of chl-*a*, a number of algorithms have been developed to estimate chl-*a* concentration from remotely-sensed reflectance data (ocean colour) for the surface ocean (typically representative of approximately the top 10 m of the water column, but depth varies with water clarity [33]). These methods include empirical relationships, semi-analytical models and neural networks [34]. Satellite-based monitoring of ocean colour now provides over two decades of continuous observation, capturing the seasonal cycles of surface phytoplankton at the global scale [35,36].

These observations of chl-*a* concentration have shown that it is typically higher in coastal regions, where phytoplankton have access to higher nutrient concentrations at the land–sea interface, when compared with the open ocean. The coastal zone is regarded as one of the most valuable yet vulnerable habitats on Earth [37]. It has the richest biodiversity (across taxa) of any marine habitat [38] and is more economically valuable per unit area than open-ocean or terrestrial ecosystems [39–41]. As a consequence of higher phytoplankton biomass, the coastal zone provides a significant proportion of global fish catch [42].

Despite the diverse array of measuring platforms available, monitoring phytoplankton in nearshore coastal regions (e.g., surf zone) is challenging and seldom documented in the literature. The dynamic nature of coastal nearshore regions makes the use of platforms such as ocean robotics, ships and moorings difficult, as they risk significant damage by crashing waves, shallow water, and strong tidal currents. While the use of satellites avoids these issues, nearshore coastal waters are often masked in satellite products, owing

to the optically complex nature of coastal waters [43] and the impact of land adjacency effects [44,45]. The lack of chl-*a* data in this region highlights the need to explore and develop new phytoplankton measuring platforms for the nearshore coastal zone.

One solution to a lack of coastal data is through citizen science, tapping into the vast numbers of citizens who regularly immerse themselves in nearshore coastal waters [46]. For example, it has been estimated that 40 million measurements on environmental indicators per year could be acquired by surfers in the UK [47]. Sea surface temperature (SST) data collected by surfers have already proven useful for evaluating satellite SST data [48,49]. The development of the Smartfin—a surfboard fin with an integrated environmental sensor package [50,51]—has lead to the automation of citizen science data collection by surfers and has proven to provide accurate measurements of SST [52,53]. Other similar initiatives for nearshore data collection include the use of divers [54–56], stand-up paddle boards [57], kayaks [58,59] and recreational sail boats [60]. The majority, however, have focused on collecting physical or chemical variables, rather than information on phytoplankton. While other citizen science initiatives have studied phytoplankton using Secchi disks and colour scales [61–65], none have seen the development of a Smartfin-like platform for automatically monitoring phytoplankton in the nearshore.

Here, we take a step in that direction. We investigate whether discrete measurements of chl-*a* collected by a surfer could be useful for understanding phytoplankton dynamics in the nearshore, and if so, motivate the development of new technology required to do this routinely. Whereas nearshore phytoplankton have been the subject of intense research (e.g., [66–69]), relatively few studies have compared seasonal dynamics in phytoplankton between nearshore and offshore coastal waters (e.g., [70–73]). To bridge this literature gap, we compared seasonal data on chl-*a* and other physical-chemical variables collected at an offshore and nearshore coastal locality, the latter by a surfer, with the aim to elicit a further understanding on the seasonal controls of phytoplankton biomass in coastal waters. To address this aim, the following three research questions were addressed:

1. Is the seasonal cycle of chl-*a* significantly different between a nearshore and offshore coastal location in Plymouth, UK?
2. Is the relationship between chl-*a* and the physical environment significantly different between these two nearshore and offshore locations?
3. Is there a difference in how phytoplankton are limited at these two nearshore and offshore locations?

## 2. Methodology

### 2.1. Study Area

The sampling locations used in this study are situated in the nearshore and offshore coastal regions of southwest England, near to the city of Plymouth (Figure 1). Bovisand Beach (latitude = 50.335° N, longitude = −4.122° E) represents the nearshore region and Station L4—situated within the Western Channel Observatory (WCO) approximately 7 km offshore (11 km from Bovisand Beach) (latitude = 50.250° N, longitude = −4.217° E)— represents the offshore coastal region.

The WCO, situated within the western English Channel (www.westernchannelobservatory.org.uk; accessed on 2nd May 2021), includes two oceanographic moorings at Station L4 and E1 [74,75]. The Station L4 location was chosen as the coastal offshore region in this study for two reasons. Firstly, hourly measurements of key variables are available from the L4 buoy, and weekly vertical profiles of key variables collected from the R/V Plymouth Quest are available at the station [76]. Secondly, the same *in-vitro* fluorometric method was used to measure chl-*a* at L4 and Bovisand, thus allowing for a common comparison of chl-*a* data between the two locations. Bovisand Beach was chosen as the nearshore coastal region based on: (i) geographical proximity to Station L4; (ii) proximity to Plymouth Marine Laboratory (PML), allowing for timely transfer of samples; and (iii) the presence of a trained scientist (also a surfer) who lived in close proximity to the beach and was willing to collect data.

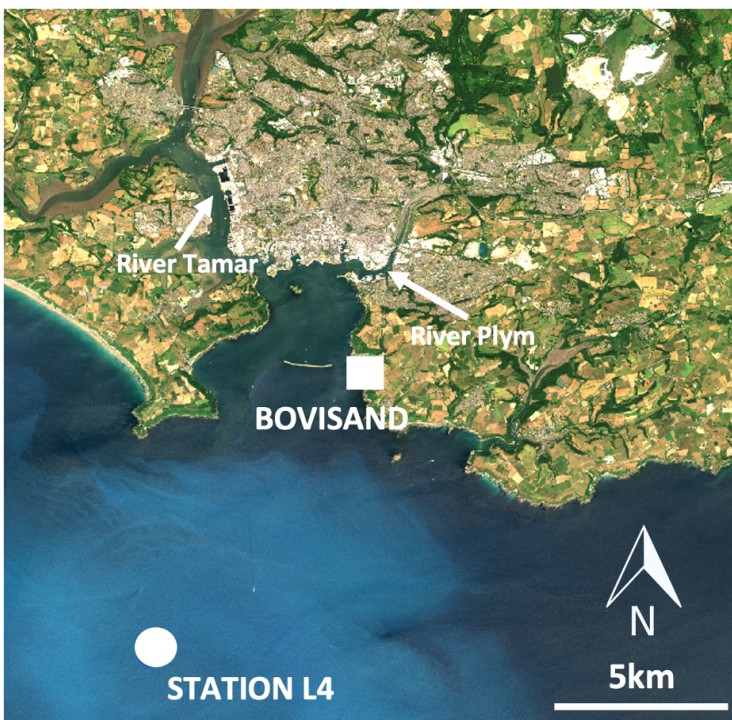

**Figure 1.** Study site map showing nearshore (Bovisand Beach), and offshore (Station L4) coastal locations sampled in this study. Image taken by Sentinel-2 satellite on 29 June 2018. Chalky white appearance in water offshore suggests the presense of coccolithophores.

### 2.2. Data Acquisition

To address the three research questions, data from both locations on phytoplankton biomass (chl-*a* concentrations), the physical environment (SST and light availability) and, where available, surface nutrients were required. Hereafter, we refer to "Station L4" as data collected during routine sampling of the WCO using a CTD profiling rosette aboard the R/V Plymouth Quest, the "L4 buoy" as data collected using the L4 autonomous buoy, and "L4" as reference to the general location within the WCO.

#### 2.2.1. Bovisand *In-Situ* Data
Sample Collection

A total of 67 corresponding chl-*a* and SST measurements were collected at Bovisand between September 2017 and September 2018. In total, 69 chl-*a* samples were initially collected but 2 were omitted: 1 sample was removed when a filter pad was damaged post-filtering, and the other was removed due to an unusually high turbid outflow from the local stream at Bovisand causing an anomalous chl-*a* reading. Sample collection was adapted from the methodology proposed by the United States Environmental Protection Agency [77].

At the end of a surfing session, a 660 mL water sample was collected in a sealable polycarbonate bottle (Figure 2B) after being thoroughly purged with seawater to remove potential contaminants. A 660 mL sample allowed for duplicate samples, with the purpose of quality control and to permit chl-*a* uncertainty to be calculated (see Section 2.2.4). The sample was then immediately taken back to the surfer's home and filtered. First, the filtration kit (Figure 2A) was purged with a small amount of the collected water sample to reduce the risk of cross-contamination with previously prepared samples. Whatman 47 mm (nominal size 0.7 μm) GF/F microfibre filters (Figure 2E) were used in the filtration rig, owing to the suitability of these filters in studying phytoplankton in coastal waters [78,79]. Before filtration, the water sample was gently rotated to ensure phytoplankton were in suspension. Half (~300 mL) of the water sample was then carefully poured into the headspace (Figure 2A) and filtered using a hand pump (Figure 2F). Pumping pressure was maintained

below 5 psi to ensure phytoplankton and filters were not damaged [77,80]. Each filter was then carefully removed and folded using tweezers and gloves to reduce the risk of contamination, placed in a labelled polycarbonate vial, and sealed. Each water sample yielded duplicate filters (two ∼300 mL samples) with the exception of three samples, where duplicates were not possible. Vials were then immediately placed into a −18 °C freezer. Within typically 24 h, vials were transported in an iced cool bag (∼30 min transport time) to PML and placed in a −80 °C freezer to preserve the pigment before analysis [81].

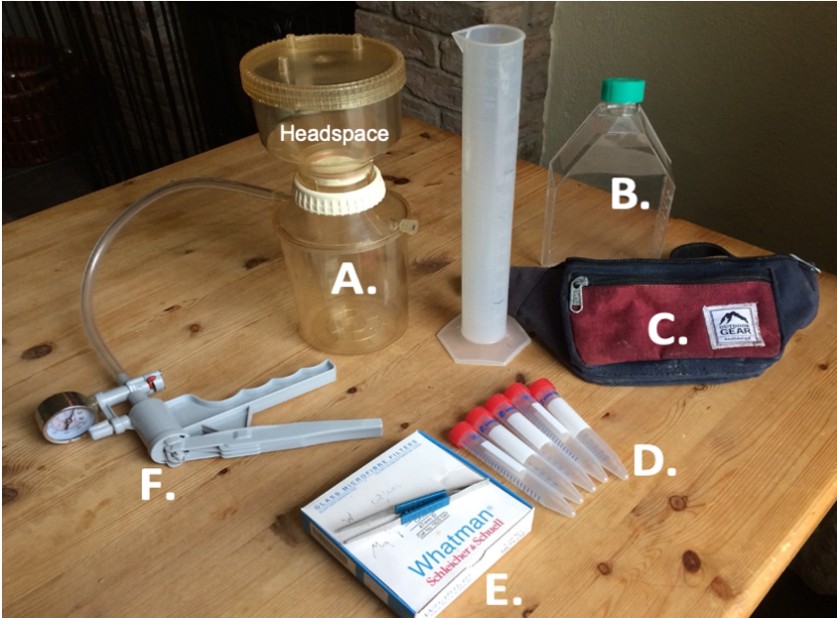

**Figure 2.** Kit used for preparation of Bovisand samples: filtration kit (**A**), sample bottle (**B**) carried into the surf using a bumbag (**C**), polycarbonate vials (**D**), Whatman 47 mm 0.7 μm GF/F microfibre filters (**E**) and a hand pump (**F**) that was used to facilitate the filtering process.

Sample Preparation and Fluorometric Analysis

To prepare the samples for analysis, the vials had 10–20 mL of 90% acetone added to them, ensuring the filter was completely submerged. The vials were then refrigerated at ∼4 °C for 24 h [82]. This allowed the chl-*a* pigment from the filters to dissolve into solution, which was then taken for fluorometric analysis.

*In-vitro* fluorometric analysis was carried out according to Welschmeyer [82] using a calibrated Turner Designs 10AU Benchtop and Field fluorometer or a Turner Trilogy 7200-000 fluorometer. Prior to analysis, the Turner instrument was calibrated with stock solutions using Sigma-Aldrich 1 mg Pure chl-*a* standard (C6144-1MG). The exact absorptions of each stock solution were recorded using a spectrophotometer which were then run through the Turner fluorometer; generating a calibration curve that was used to determine the chl-*a* concentrations of the collected water samples. Each of the prepared Bovisand solutions were then carefully transferred into a cuvette and run through the fluorometer. This yielded raw fluorescence values which were converted to a chl-*a* concentrations using the slope of the calibration curve and knowing the volumes of the samples filtered and the acetone used in the pigment extractions.

Between each sample, the cuvette was rinsed with acetone and thoroughly dried to reduce the risk of cross-contamination. Fluorometric analysis was repeated for all duplicates over a series of three sessions, calibrating the fluorometer in the same way at the beginning of each session. These data are openly available through the British Oceanographic Data Centre [83].

Sea Surface Temperature (SST) at the Beach

The median SST during each surfing session was primarily collected using a Smartfin (Figure 3A), a surfboard fin with an integrated temperature sensor [50,51], following the methods described by Brewin et al. [52]. There were two cases where SST data were collected using the TidbiT v2 sensor (with protective boot, see Figure 3B) attached to the surfboard leash when the Smartfin failed to record SST [52]. The mean absolute deviation and mean bias between these two methods in UK waters were found to be 0.03 and 0.00 °C, respectively [52], justifying the combination of methods. Following Vanhellemont et al. [49], the uncertainty of SST measurements was approximated by taking the square root of the sum of the squared calibration error of the Smartfin (set to 0.05 °C based on laboratory comparisons of sensors [52]) and the squared median absolute deviation in temperature during each surfing session. These data are openly available through the British Oceanographic Data Centre [83].

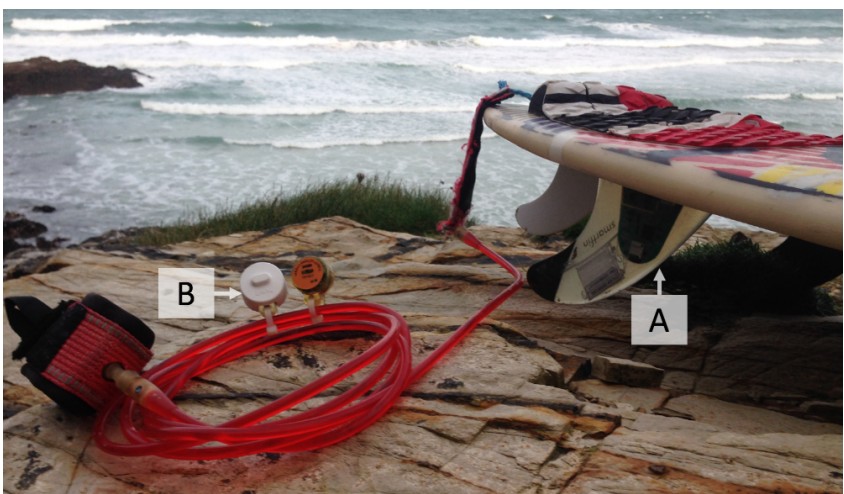

**Figure 3.** Image showing Smartfin (**A**) and TidbiT v2 temperature sensor with protective boot (**B**).

### 2.2.2. Station L4 *In-Situ* Data

A total of 41 corresponding chl-*a* concentrations (*in-vitro* fluorescence) and SST measurements were collected using the CTD profiling rosette aboard the R/V Plymouth Quest, as part of PML's weekly sampling of L4. These data were extracted from the WCO data repository [84]. As with the Bovisand samples, the data run from September 2017 to September 2018, capturing the seasonal cycle of chl-*a* and the physical environment at L4. Surface chl-*a* was measured according to Welschmeyer [82], using almost identical methods to those used for the Bovisand samples. The main difference was that 100 mL of seawater was filtered through a smaller (25 mm) GF/F filter in triplicate. SST from Station L4 is an average of the top five metres [84] and was measured with a SeaBird SBE 19+, with an uncertainty of 0.005 °C [85]. SST for one station (on the 10 April 2017) was not available.

### 2.2.3. Auxiliary Data
L4 Autonomous Buoy

Surface chl-*a* and SST recorded by the L4 autonomous buoy [86] were used to assist the description of the seasonal cycles presented here but were not considered for formal analysis, owing to the use of the less accurate and potentially more problematic (e.g., from non-photochemical quenching) *in-vivo* fluorescence method used to measure surface chl-*a* [87,88]. Additionally, data were only available for the early and latter months of the period of study, as the buoy was taken out of the water late October 2017 for essential maintenance and was not in operation again until late May 2018.

Surface Nutrients

Surface nitrate, nitrite, phosphate, and silicate were extracted from the WCO data repository [89]. Clean seawater samples were collected from the R/V Plymouth Quest and returned in the cool and dark, and as soon as possible, to PML where triplicate samples were analysed using a 5-channel Bran and Luebbe segmented flow colorimetric autoanalyser. Quality control procedures were carried out using KANSO certified reference materials. Surface nutrient data were used to help understand the seasonal cycles of phytoplankton at L4. Surface nutrients were not available at Bovisand Beach.

Stratification

To assess stratification at L4, density and temperature data were extracted from the WCO data repository [84] and used to create a stratification index. The data available for 5 m upwards and 45 m and below were averaged and the difference between these averages was calculated and used as a proxy for stratification.

Microscopy Data

Surface phytoplankton microscopy data were available at Station L4 [90]. Samples were collected from a 10 m depth using a 10 L Niskin bottle, whereby paired 200 mL water samples were fixed in acid Lugol's iodine [91] for enumerating phytoplankton cells >2 μm and neutral formaldehyde for enumerating coccolithophores. Samples were returned to PML and stored in cool, dark conditions until analysis using light microscopy and the Utermohl counting technique [92] following the British and European standards [93]. Samples were settled for >48 h and all cells >2 μm were enumerated, and identified to the species level (where possible) using an inverted microscope at either ×200 or ×400 magnification [90]. Cells were categorised into six groups; flagellates, diatoms, *Phaeocystis*, coccolithophorids, dinoflagellates, and ciliates [90]. Only data on coccolithophorids were used in our study, to confirm the presence of coccolithophores during the start of a stratification event.

Satellite Data

Photosynthetically Available Radiation (PAR) data for Bovisand and L4 were extracted from September 2017 to September 2018 following methods described by Brewin et al. [52], see their Section 2.4.3. PAR data consisted of daily products taken from three ocean-colour sensors (MODIS-Aqua, MODIS-Terra, and VIIRS). PAR uncertainty was estimated at 11% following Gould Jr et al. [94]. Additional chl-*a* time-series data for the period of study were extracted from the European Space Agency Ocean Colour Climate Change Initiative (OC-CCI) dataset [36]. Daily OC-CCI 1 km mapped remote-sensing reflectance data at 4 wavelengths (443, 490, 510, and 555 nm) were extracted and converted to nearsurface chl-*a* concentrations using the NASA OC4v4 algorithm [95]. The median chl-*a* concentration of the nine pixels—3 × 3 box centred over L4—for each day of the year (September 2017 to September 2018) was computed, giving a daily time series of chl-*a* at L4. Additional chl-*a* time-series data were not available for Bovisand, owing to data being masked from this location in the OC-CCI processing. The OC-CCI data were used to corroborate the L4 *in-situ* time series of chl-*a* during the period of study. A single mapped image from ESA Sentinel 2 taken on the 29 June 2018 (true colour RGB composite) was produced for the analysis, indicating the presence of a coccolithophore bloom at L4 (see Figure 1).

2.2.4. Calculating chl-*a* Uncertainty

Uncertainties in chl-*a* ($\Delta$) were approximated for each sample according to

$$\Delta = \sqrt{\sigma^2 + \delta^2},\tag{1}$$

where $\sigma$ is the standard deviation of the replicates for each sample, and $\delta$ is an estimate of the accuracy of the fluorometer. Equation (1) was designed to capture the key components

of the uncertainty budget ($\sigma$ representing uncertainty in the sampling protocol and $\delta$ the uncertainty of the instrument), but we acknowledge it does not account for all the components. Owing to the log-normal distribution of chl-*a* in the ocean [96], all components of Equation (1) were computed after $\log_{10}$ transformation. For samples where replicates were not available, $\sigma$ was taken to be the average standard deviation of samples with replicates (conducted separately for Bovisand and L4 data). $\delta$ was estimated by matching 569 co-located and concurrent estimates of chl-*a* from water samples collected at Station L4 between 1999 and 2013 using the Turner Designs 10AU Benchtop and Field fluorometer and HPLC. The $\log_{10}$-transformed chl-*a* concentrations were highly correlated ($r = 0.82$), with no systematic difference (mean difference = 0) and an unbiased root mean squared error of 0.19, which was assigned the $\delta$ term in Equation (1). It is worth noting that calculating the uncertainty in this way assumed that the HPLC data were representative of the actual chl-*a* concentration, acknowledging that the HPLC method has an inherent level of uncertainty [97].

### 2.2.5. Statistical Analysis

All statistical analyses and plotting were carried out in R Studio (version 1.3.1093). Following formal and graphical tests of normality, the non-parametric Spearman's rank correlation coefficient was used when correlating chl-*a* concentration and the physical variables (SST and PAR).

## 3. Results

### *3.1. Seasonal Cycle of chl-a*

Figure 4 describes the seasonal cycle of chl-*a* as observed in the measurements collected *in-situ* at Bovisand, Station L4 (CTD data) and by the L4 buoy. Satellite data from OC-CCI for L4 are also shown in Figure 4A as a way of corroborating the trends shown in the L4 *in-situ* data. Though in good agreement, the satellite data are patchy, illustrating the importance of the *in-situ* time series for quantifying seasonality. The monthly binned averages are presented in Figure 4B to highlight the agreements between data collected *in-situ* at Bovisand and Station L4, and to confirm if any significant differences between the two locations exists.

The seasonal cycle of chl-*a* captured at Bovisand and L4 shows a steady decline in chl-*a* concentrations throughout autumn and winter (Figure 4). At the start of spring, chl-*a* concentrations increase before declining at the beginning of summer and begin to recover come autumn. This seasonal cycle is typical of North Atlantic phytoplankton [98]. Through the autumn and winter months of 2017, chl-*a* concentrations at the two locations show a good nearshore–offshore agreement, as shown by the many overlapping data points during this period (Figure 4A). This agreement begins to deviate during the summer period highlighted in yellow; chl-*a* concentrations at L4 are shown to drop during this period, whilst Bovisand chl-*a* remains high. The nearshore–offshore agreement then begins to improve post-summer. The monthly binned averages of chl-*a* in Figure 4B show the change in nearshore–offshore agreement more clearly and indicate that chl-*a* is significantly higher at Bovisand during July and August than at L4 (indicated given error bars do not overlap).

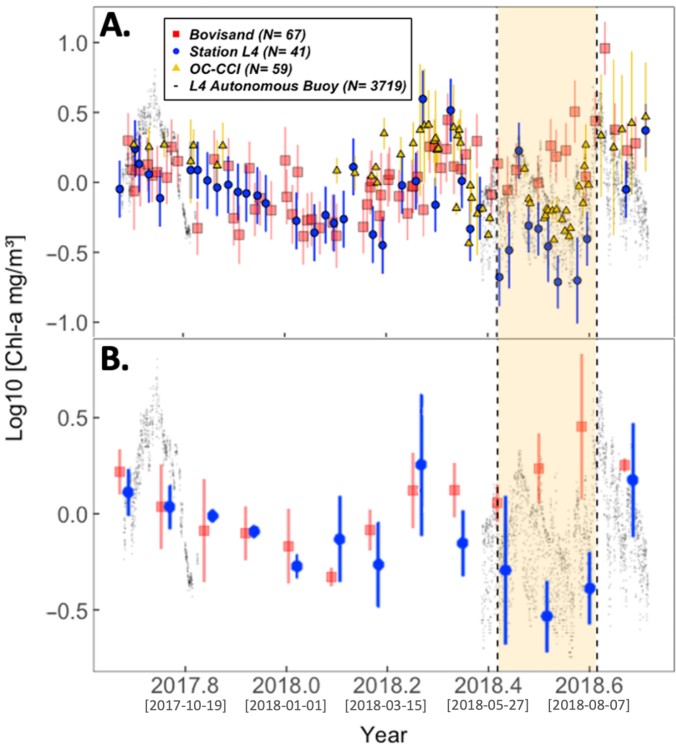

**Figure 4.** Seasonal cycles of chl-*a* captured at Bovisand and L4. (**A**) is the chl-*a* data for both Bovisand and L4 from various sources: Bovisand *in-situ* data are shown as red squares; L4 *in-situ* data are shown as blue circles (Station L4 CTD data) and black dots (L4 autonomous buoy). Auxiliary satellite OC-CCI chl-*a* data at L4 are shown as yellow triangles. (**B**) Monthly binned averages of Bovisand *in-situ* data and Station L4 *in-situ* data with L4 autonomous buoy data underlain to assist in visualising the seasonal cycle. In (**A**), the vertical bars for Bovisand and Station L4 data represent the measurement uncertainty (Section 2.2.4) and the standard deviations of the OC-CCI data. In (**B**), the vertical bars are the standard deviation of the monthly bins with the exception of the August bin at L4, which only consisted of one value and so represents the uncertainty of that measurement. The yellow shaded area represents the summer period June to August. All chl-*a* data are $\log_{10}$ transformed. Square bracketed dates on the x-axis ticks represent Year-Month-Day.

### 3.2. Physical Environment Time Series

Figures 5 and 6 present the time series of SST and PAR for both Bovisand and Station L4 using all available data at both locations. The yellow shaded area indicates the same summer period highlighted in Figure 4 (June–August). For both Figures 5 and 6, subplots B present monthly binned averages of the data in subplots A, with the purpose of clarifying the monthly trends seen in the raw data and to see if any significant different exists between the two locations. Across both locations, SST decreases at approximately the same rate until March, reaching lows of 6.8 and 7.8 °C for Bovisand and Station L4, respectively (Figure 5). Following March, SST is shown to increase and peaks in July to August for both locations. SST reaches highs of 17.7 °C at Bovisand and 18.9 °C at Station L4 during this time. After the summer peak, SST is shown to decrease during Autumn.

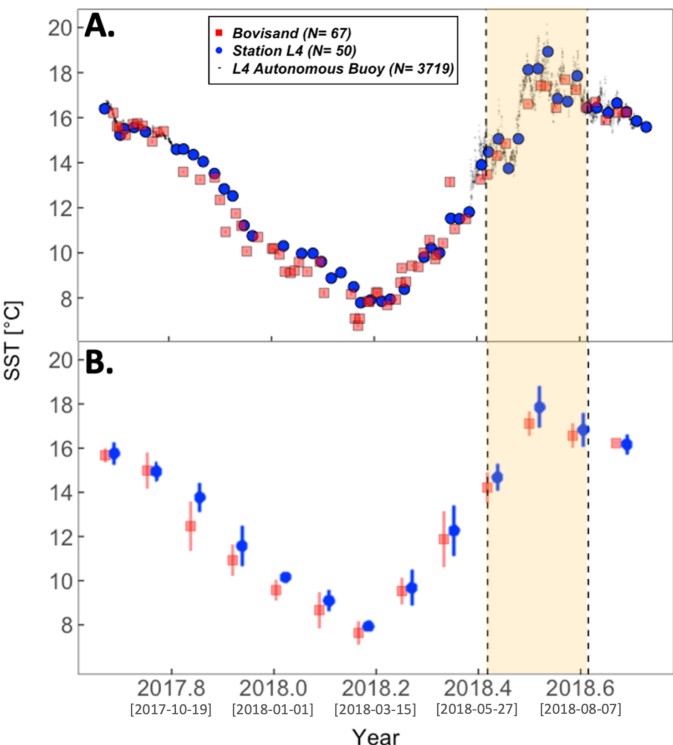

**Figure 5.** (**A**) Time series of SST from Bovisand *in-situ* data (red squares) and Station L4 data (blue circles). SST data from the L4 autonomous buoy are underlain in (**A**) (black dots) as a way of verifying the trends seen in the *in-situ* Bovisand and Station L4 data. (**B**) is the monthly binned averages of the Bovisand and Station L4 *in-situ* SST data in (**A**). Vertical bars in (**A**) represent the measurement uncertainty and in (**B**), represent the standard deviation of the bins. The yellow shaded area represents the summer period June to August. Data for Station L4 were available from the WCO data repository [84]. Square bracketed dates on the x-axis ticks represent Year-Month-Day.

During the same period, PAR across both locations decreased into the winter, reaching its lowest point around the winter solstice, before rapidly increasing again following the advent of spring (Figure 6). PAR then reached its peak at both locations around the summer solstice, and begins to decrease as autumn approaches. No significant differences in SST and PAR between the two locations are observed (Figures 5B and 6B) in the monthly averages.

Coinciding with the same summer period (June–August)—highlighted in yellow in Figures 5–7—was a strong stratification event at L4, as shown in the temperature and density differences between the surface and 50 m in Figure 7. A coccolithophore bloom observed by the chalky white appearance in Figure 1 on 28 June 2018, and confirmed with L4 microscopy data (3772 coccolithophorid cells per mL on 2 July 2018 (highest coccolithophorid abundance over the study period), dominated entirely by *Emiliania huxleyi*), further confirms the presence of stratification during this period, as this phytoplankton group are known to be strongly associated with stratified waters [99]. Coccolithophorid abundances for all other samples during the annual cycle were less than 500 cells per mL.

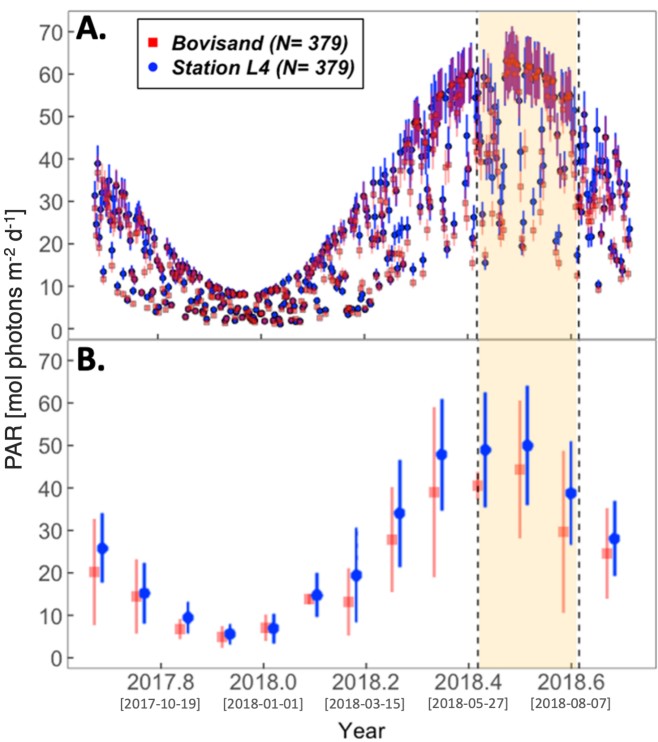

**Figure 6.** (**A**) Daily time series of satellite PAR at Bovisand (red squares) and L4 (blue circles). (**B**) is the monthly binned averages of the satellite data in (**A**). The vertical bars in (**A**) represent the measurement uncertainty of the satellite sensor and in (**B**), represent the standard deviations of the monthly bins. The yellow shaded area represents the summer period June to August. Square bracketed dates on the x-axis ticks represent Year-Month-Day.

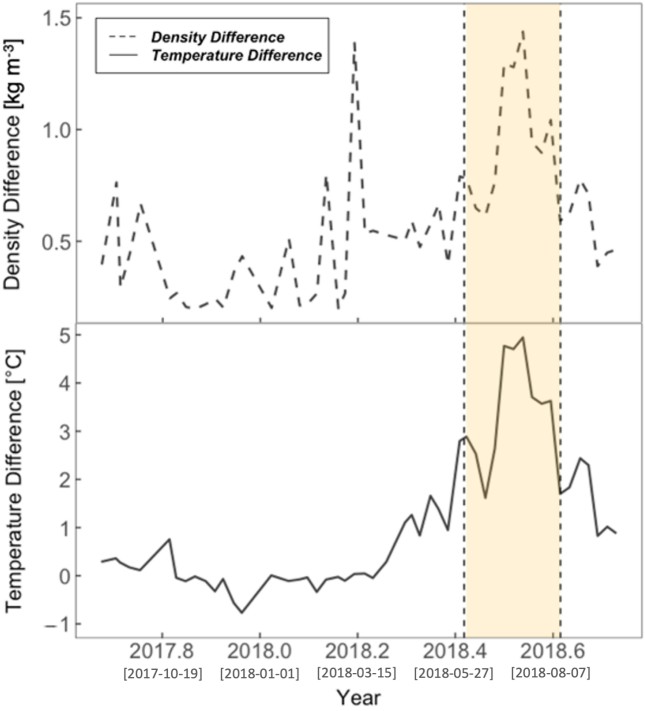

**Figure 7.** Density and temperature difference between the surface (0 to 5 m) and bottom (45 to 50 m) at L4, used as a stratification index. High stratification is indicated by a higher difference value. The yellow shaded area represents the summer period June to August. Square bracketed dates on the x-axis ticks represent Year-Month-Day.

The surface nutrients recorded at L4 (Figure 8) are generally shown to increase during the autumn and winter periods before rapidly crashing during the spring. This trend is different across the four nutrients described in Figure 8, with nitrite reaching its peak and then rapidly depleting earlier than the other three nutrients. The similarity across all nutrients is: (i) depletion following peak concentrations, and (ii) the lowest concentrations being observed during the summer period (Figure 8, yellow shaded area). Coinciding with the peaks in SST (Figure 5), PAR (Figure 6), and stratification (Figure 7), the lowest surface nutrient concentrations are observed at L4 (Figure 8).

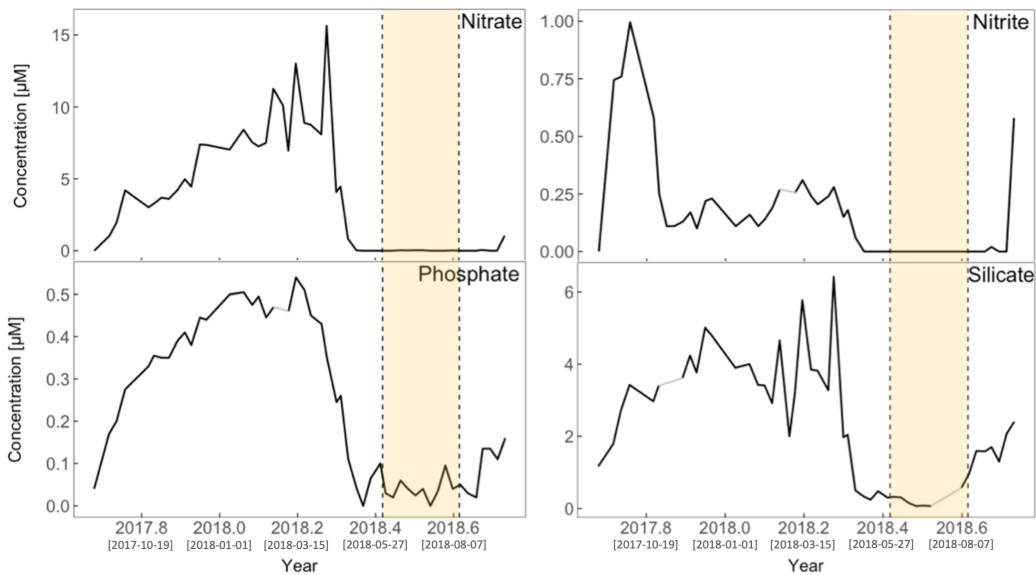

**Figure 8.** Surface nutrient concentrations at L4 over the study period. Lighter shades (grey) represent interpolation for instances where data were unavailable. The yellow shaded area represents the summer period June to August. Square bracketed dates on the x-axis ticks represent Year-Month-Day.

### 3.3. Relationships between chl-a and the Physical Environment

The results of Spearman's rank correlation between chl-*a* and the physical variables (SST and PAR) at both Bovisand and Station L4 are presented in Table 1 and Figure 9. At Bovisand, a significant positive correlation between chl-*a* and SST was observed ($r = 0.56$, $p = < 0.001$), whilst at L4 the correlation between chl-*a* and SST was not significant ($r = -0.05$, $p = 0.75$). The correlation between chl-*a* and PAR at Bovisand was significant and positive ($r = 0.42$, $p = < 0.001$) but these variables were not significantly correlated at L4 ($r = -0.21$, $p = 0.20$). Interestingly, when the summer months June, July and August are omitted from L4 correlations—which is when agreement in chl-*a* concentrations between L4 and Bovisand begin to deviate (Figure 4)—the correlation coefficients change considerably, from negative to positive (Table 1). This is especially the case for chl-*a* and SST at L4, which becomes significantly positively correlated ($r = 0.52$, $p = 0.004$).

**Table 1.** Results from correlation coefficient analysis (Spearman's) between chl-*a* and the physical variables (SST and PAR), including omission of summer months represented by the yellow shaded area in Figures 4–8.

| | Location | chl-*a* ($\log_{10}$) and SST [$] | chl-*a* ($\log_{10}$) and PAR [$] |
|---|---|---|---|
| | Bovisand | $r = 0.56$, $p < 0.001$, $N = 67$ | $r = 0.42$, $p < 0.001$, $N = 67$ |
| Station L4 | All data | $r = -0.05$, $p = 0.750$, $N = 40$ | $r = -0.21$, $p = 0.200$, $N = 41$ |
| Station L4 | June/July/August omitted | $r = 0.52$, $p = 0.004$, $N = 29$ | $r = 0.31$, $p = 0.100$, $N = 30$ |

[$] $r$ = Spearman's correlation, $p$ = significance level, $N$ = number of measurements.

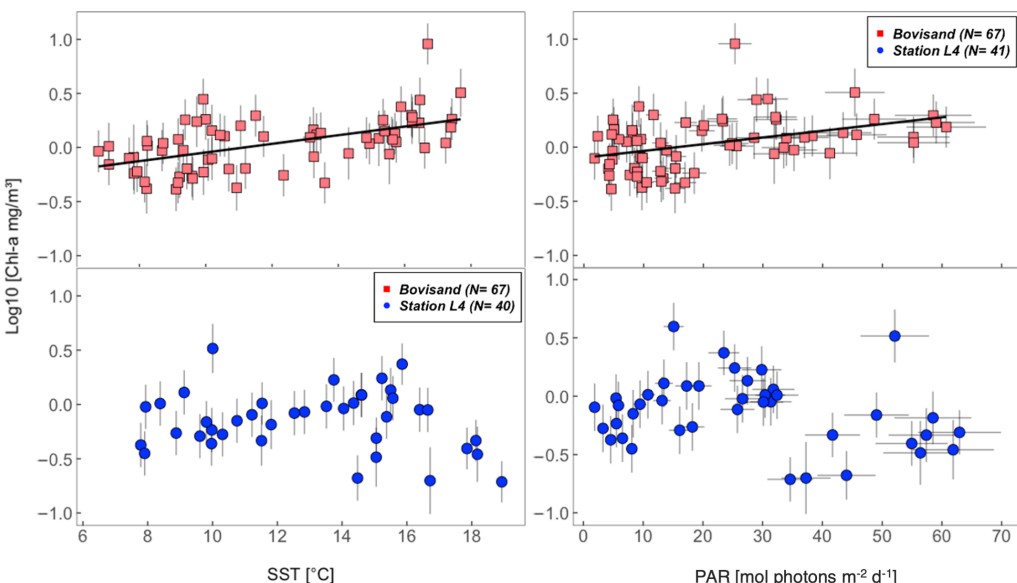

**Figure 9.** Scatter plots of chl-*a* against physical variables (SST and PAR) at Bovisand (red squares) and Station L4 (blue circles). Error bars (vertical and horizontal) represent the measurement uncertainty for both the explanatory (SST and PAR) and response (chl-*a*) variables. Ordinary least squares (OLS) regression line is not plotted in the L4 plots due to their non-significance (Table 1: *p*-values > 0.05).

## 4. Discussion

### 4.1. Research Questions

Our results show a statistically significant difference in chl-*a* concentrations between the two coastal locations studied (nearshore and offshore) during the summer. In both July and August, chl-*a* concentrations at Bovisand are significantly higher than at L4 (Figure 4B). The correlation coefficients presented in Table 1 and Figure 9 suggest that both SST and PAR are significantly positively correlated with chl-*a* at Bovisand, with no significant correlation existing in the Station L4 data (except when summer months are removed). During the summer period (June–August), strong stratification is observed in the density and temperature differences at L4 (Figure 7), which coincides with the depletion of various essential nutrients (Figure 8). Following the omission of the summer months in the Station L4 seasonal cycle of chl-*a*, the change from a negative to a positive correlation between chl-*a* and PAR suggests that it is not light that is limiting phytoplankton growth at L4 during the summer but that nutrients are the limiting factor. Although surface nutrient data were not available at Bovisand, higher chl-*a* remained over the summer period (likely due to proximity to land and rivers, see Figure 1), suggesting that nutrients were not limiting at Bovisand, and light was still controlling phytoplankton biomass (positive correlation with PAR).

### 4.2. Nearshore-Offshore: Why the Difference?

The notable period of interest in this study is the summer months of July and August. This is where chl-*a* concentrations at Bovisand (coastal nearshore) and L4 (coastal offshore) become significantly different from each other (Figure 4B). Up until this point in the seasonal cycle, chl-*a* concentrations recorded at both locations show a decline during the winter months and subsequent bloom at the advent of spring. This seasonal cycle is typical of North Atlantic phytoplankton which has been observed in numerous studies [100–102]. Low light levels during the winter limit phytoplankton photosynthesis in the mixed layer [103–105]. The shoaling of the mixed layer and rapidly increasing light levels during spring (Figure 6)—understood to be the principal trigger of the spring bloom—see rapid increases in phytoplankton biomass and consequential crashes in nu-

trient concentrations [15]. In both locations, this explanation is consistent with trends observed in Figure 4, showing the spring bloom around the month of April.

Moving into the summer, higher surface temperatures (Figure 5) drive stratification within the euphotic zone, with other influences such as low wind and salinity also playing a role [106,107]. Such stratification is observed when large temperature differences exist within the vertical water column [16] (Figure 7). During summer stratification, it is understood that nutrients become the limiting factor controlling phytoplankton growth in the mixed layer [98,100] due to the lack of vertical mixing in the water column, which can replenish nutrients [106]. The results from this study suggest that this is what is happening at L4 (Figures 7 and 8, Table 1), and results in a drop in offshore phytoplankton biomass (chl-*a* concentrations) (Figure 4: blue circles). When removing the nutrient-depleted summer monthly data, the change from negative to positive correlations between chl-*a* and PAR at L4 in Table 1 further back up this claim. Unfortunately, data on surface nutrient concentrations at the coastal nearshore location (Bovisand) were not available. However, it is widely understood that the interface between the land and the sea is responsible for a significant influx of nutrients (enhanced from human derived anthropogenic influences, for example agricultural and industrial inputs of nutrients) into the coastal nearshore [108–110], which has been known to influence phytoplankton growth dynamics [111,112]. The increased (or continual) supply of nutrients in the nearshore (from land sources, mixing of the water column by wave action, and even processes such as increased remineralization in sediments in response to warming of organic matter), suggests that, unlike phytoplankton at L4, growth is not limited by nutrients at Bovisand but is instead controlled by light all year round (Figure 9, Table 1: chl-*a* and PAR positive correlation).

### 4.3. Implications for Understanding Coastal Phytoplankton Dynamics

Understanding phytoplankton dynamics in the coastal environment is especially important owing to the value of ecosystems services in this area and their vulnerability to climate change [37]. This places value on the *in-situ* data collected in very nearshore waters, where OC-CCI satellite data are not available, and where it is a challenging region to sample using conventional *in-situ* methods. Our understanding of how phytoplankton will respond to climate change is still widely debated [113]. Nonetheless, it is generally thought that the oceans will become more stratified with warming global temperatures [114]. Increased stratification is suggested to reduce dilution of active zooplankton and thus increase grazing pressure [115], and exacerbate nutrient limitation in surface waters, both of which may lead to reduced phytoplankton productivity and biomass [107]. Warming temperatures are also suggested to change phytoplankton assemblage composition and increase the prevalence of harmful algal blooms [116].

The findings of this study, particularly the differences in phytoplankton chl-*a* concentration during summer stratification between coastal nearshore and offshore environments, may have important implications for understanding how coastal phytoplankton dynamics are changing. For example, under a scenario of enhanced coastal summer stratification, differences in chl-*a* concentration may become greater between nearshore and offshore environments at our study site. However, the nutrients that nearshore phytoplankton have access to may themselves be subject to other factors impacted by humans and climate change. Nonetheless, addressing questions on the impact of climate change on coastal phytoplankton ultimately requires time-series data on phytoplankton far longer than that used in our study [36,117]. It maybe that surfers and recreational ocean users could help build such datasets for future studies.

### 4.4. Limitations of Our Study

Whereas chl-*a* seasonality at L4 and Bovisand was quantified using a common and consistent approach (*in-vitro* fluorometric method, see Section 2.2), with consideration of measurement uncertainties, there were minor differences in how the data were processed (e.g., different filtration rigs, filtration volumes, plastic carboy bottles used, and freezing

procedure). Future work should aim to quantify if these minor differences cause any systematic changes in data collection and analysis. Future work should also aim to collect nutrient data in the nearshore. Without this data, it was difficult to answer our third research question "Is there a difference in how phytoplankton are limited at these two nearshore and offshore locations?" Analysis of nutrient stoichiometry between nearshore and offshore coastal environments could also shed light on the differences in chl-*a* observed between locations during the summer. Additionally, direct experiments quantifying limitation by light or nutrients on these nearshore and offshore phytoplankton communities should be explored, perhaps under laboratory settings. This would help to quantify whether the nutrient depletion observed at Station L4 (Figure 8) leads directly to nutrient limitation.

The temporal frequency of sampling conducted in this study was sufficient to capture the general seasonal cycles of chl-*a* in nearshore and offshore locations, but insufficient to investigate differences at finer temporal scales. For that, one may require simultaneous data collection at higher frequency (e.g., hourly to daily) than the near weekly sampling (with $N = 67$ for Bovisand and $N = 41$ for Station L4) conducted here, and likely a different approach. Augmenting tools such as Smartfins with autonomous chl-*a* sensors (e.g., *in-vivo* fluorometers) and tapping into a larger array of recreational platforms [46] could be a way of doing this.

Considering the high spatial variability in nearshore coastal waters, it is important to also recognise that the findings observed at Bovisand may not be representative of other nearshore coastal waters in the southwest, UK, and beyond. Bovisand is located near to the Plymouth Hoe, where two rivers (the Tamar and the Plym, see Figure 1) meet, and consequently may be subject to a different physio-chemical environment than many other beaches in the southwest. Bovisand is relatively exposed to southwest swells, and therefore likely a higher energy environment than some more sheltered beaches along the southwest coast of the UK, again which could impact the physio-chemical environment. Whereas this may limit the generality of our findings, it underlines major gaps in nearshore coastal observations.

Notwithstanding these limitations in spatial-temporal sampling, there are not many coastal stations in the UK and worldwide like L4, with systematic weekly long-term sampling, and to our knowledge, very few that have been sampled annually alongside a nearshore station (Bovisand), for addressing questions such as those posed in our work, on the phenology of phytoplankton. Further development of monitoring platforms in the coastal zone is required to adequately fill in the gaps in data availability identified.

*4.5. Future Directions*

It is known that a deep chlorophyll maximum (DCM) often develops following summer stratification in the North Atlantic [118,119], and at Station L4 [84]. Whilst this study focussed exclusively on surface chl-*a*, future research should also focus of understanding vertical phytoplankton dynamics in relation to changes in the physical and chemical environment [120,121]. Whereas this study has focused on phytoplankton biomass, the composition of phytoplankton at the two sites may vary, with implications for coastal biogeochemical cycles [122]. Future research could quantify if any differences in phytoplankton composition exist between sites. Additionally, quantifying if differences in top-down control (zooplankton) exist between sites may further aid understanding of nearshore plankton seasonality.

Our findings demonstrate that chl-*a* in the nearshore environment can be significantly different than in offshore coastal waters during the summer months in the southwest, UK. Considering the important role phytoplankton play in nearshore waters, as primary producers, these results support the development of new tools and platforms capable of monitoring phytoplankton in nearshore waters at a higher spatial and temporal frequency. One solution is through the development of more automated citizen science methods of collecting phytoplankton biomass data (chl-*a* concentrations), using tools such as the Smartfin. Low-cost fluorometers are becoming smaller and increasingly accessible [26], and have been attached

to marine animals (e.g., seals) to study the vertical distributions of phytoplankton biomass in polar and sub-polar waters [123–125]. Yet, interpreting *in-vivo* fluorescence signals is challenging [88,126], especially in the surf zone [127,128]. Miniaturising other more accurate *in-situ* techniques for determining chl-*a* concentration (e.g., absorption or beam transmission measurements) could offer an alternative route [16,25,30,129]. Surfers are just one group of a whole host of potential citizen scientists that could help expand nearshore aquatic observations. Sensor integration into other recreational activities such as fishing, kayaking, scuba diving and motorboating could expand measurements of variables such as chl-*a* concentrations in both fresh and salt water environments (see Table 1 of Brewin et al. [46]). Since many of these activities occur in different conditions (e.g., calm seas), this would also tackle biases in data collection that will inevitably occur for conditions and locations preferable for surfing.

The spatial resolution and spectral capabilities of satellite-based sensors is increasing. Alongside algorithmic developments this is leading to improved correction schemes and a push to process (rather than mask) data closer and closer to the coastline. Remote sensing data within a few 100 m of the coastline will need validation, and any *in-situ* observations that can be acquired here would be of great use for satellite validation purposes [49].

Though our work shows promise in using surfers and other recreational platforms for coastal monitoring, it must be acknowledged that this comprises only a small portion of a potential wide-ranging coastal monitoring observatory. Our best picture of how the coastal ocean is changing will be found by integrating these datasets with other platforms, including ocean robotics (e.g., Argo floats, underwater gliders, and unmanned surface vehicles), the next generation of coastal buoys and drifters, ship-based autonomy, and remote sensing platforms (drones, aircraft and satellites), and with coupled physical and biogeochemical modelling [32,130–132].

## 5. Conclusions

Using measurements of chl-*a* as an index of phytoplankton biomass, the seasonal cycles of phytoplankton (September 2017 to September 2018) at a nearshore and offshore coastal location near Plymouth, UK were investigated. Where available, data on the physical and chemical environment were used to assess controls on phytoplankton.

We found no significant difference in the seasonal cycle of phytoplankton biomass between the nearshore and offshore locations from September 2017 to June 2018, which was also consistent with knowledge of the North Atlantic seasonal cycle observed in the literature. However, during July and August 2018, significant differences in chl-*a* were observed between sites, with a sudden drop in chl-*a* concentrations at the offshore location, whilst chl-*a* remained high at the nearshore location.

Data on SST and PAR at both locations were used to investigate the relationship between chl-*a* and the physical environment. We found SST and PAR to be significantly positively correlated with chl-*a* in the nearshore over the seasonal cycle, but no significant correlations were observed at the offshore location. However, once the summer months of June through to August 2018 were removed from the offshore data, the correlation coefficients were more consistent with those observed in the nearshore.

During the summer months, surface nutrient data at the offshore location showed low concentrations, suggesting that phytoplankton may be limited by nutrients. Considering chl-*a* remained high in the nearshore during this period, this led to the conclusion that phytoplankton were not limited by nutrients in the summer, possibly due to its proximity to the land and the owing to the dynamic nature of the nearshore. Unfortunately, samples for nutrient analysis were not taken at the nearshore location which meant we could not verify this conclusion.

Our work demonstrates the potential of using recreational watersports to help monitor phytoplankton dynamics in a challenging region to sample (the nearshore). Our findings highlight the possibility that climate change may impact phytoplankton in the coastal waters in different ways, depending on whether they occupy nearshore or offshore waters.

The development of a broad-ranging coastal monitoring observatory that integrates data from a wide variety of automated and citizen science-based platforms is required to better understand phytoplankton dynamics at the coastline.

**Author Contributions:** Conceptualization, E.M. and R.J.W.B.; methodology, E.M. and R.J.W.B.; software, E.M. and R.J.W.B.; validation, E.M. and R.J.W.B.; formal analysis, E.M. and R.J.W.B.; investigation, E.M. and R.J.W.B.; resources, R.J.W.B., Q.V., O.J., D.C., G.T., C.W., E.M.S.W., C.H., P.J.B., T.C. and A.J.A. data curation, R.J.W.B., Q.V., O.J., D.C., G.T., T.J., C.W., E.M.S.W. and C.H.; writing—original draft preparation, E.M. and R.J.W.B.; writing—review and editing, E.M., R.J.W.B., Q.V., O.J., D.C., G.T., T.J., C.W., E.M.S.W., P.J.B., T.C. and A.J.A. All authors have read and agreed to the published version of the manuscript.

**Funding:** R.J.W.B. is supported by a UKRI Future Leaders Fellowship (MR/V022792/1). A.A., P.B. and T.C. received financial support from the non-profit Lost Bird project. This work is supported by the UK Natural Environment Research Council's National Capability Long-term Single Centre Science Programme, Climate Linked Atlantic Sector Science, grant number NE/R015953/1, and is a contribution to Theme 1.3-Biological Dynamics. G.T., C.W., E.M.S.W. and C.H. were supported by S3-EUROHAB-Sentinel-3 products for detecting EUtROphication and Harmful Algal Bloom events (contract no. 106) from the European Regional Development Fund through the INTERREG France-Channel-England.

**Data Availability Statement:** Surfer data collected are openly available through the British Oceanographic Data Centre https://www.bodc.ac.uk/data/published_data_library/catalogue/10.5285/d6a5a863-a43d-28a9-e053-6c86abc0b1f4/). L4 datasets are available through the Western Channel Observatory (www.westernchannelobservatory.org.uk). Satellite data used are freely available through NASA (https://oceancolor.gsfc.nasa.gov) and ESA (https://climate.esa.int/en/projects/ocean-colour/).

**Acknowledgments:** We acknowledge all those involved in the collection of data used in this study. We also acknowledge all those involved with maintaining and operating the L4 buoys and the R/V Plymouth Quest. We thank Phoebe Brewin for assistance in filtering chl-*a* samples collected at Bovisand Beach. The authors thank the NERC Earth Observation Data Acquisition and Analysis Service (NEODAAS) for supplying data for this study. This work was supported by a MSci project in Environmental Sciences at the Centre for Geography and Environmental Science (CGES), College of Life and Environmental Sciences, University of Exeter, and we acknowledge all those working behind the scenes to make this programme a success, in particular Programme Directors Liam Reinhardt and Tomas Chaigneau, CGES Director James Scourse and Director of Education Michael Leyshon. We also thank Katy Sheen for comments on earlier versions of this work. We thank two anonymous reviewers for their comments on our paper.

**Conflicts of Interest:** A.A., P.B. and T.C. received financial support from the non-profit Lost Bird project. All other authors declare the absence of any commercial or financial relationships that could be construed as a potential conflict of interest.

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
