# Peer review of "On the Seasonal Dynamics of Phytoplankton Chlorophyll-a Concentration in Nearshore and Offshore Waters of Plymouth, in the English Channel: Enlisting the Help of a Surfer"

_2673-1924, doi:10.3390/oceans3020011_

Round 1
Reviewer 1 Report
This manuscript entitled "An unconventional investigation into the seasonal dynamics of phytoplankton in nearshore coastal waters of southwest England" is authored by 13 authors from 6 Institutions. The title is catchy, but it should be more informative. The basis of the study, the analyses of the chlorophyll a at two sampling stations in- and offshore Plymouth during one year, has little innovation. The result, chlorophyll a is low during the summer in the offshore station, but it remains high in the inshore station, is not unexpected. Obviously, the nearshore mixing and the inputs of the two rivers at Plymouth contribute to maintain high levels of Chl a during the summer when compared to the offshore station. If we restrict the manuscript to these contents, it has not a relevant interest. However, these are other aspects with interest. The authors used distinct methods as estimation of the Chl a: the classical analyses using a fluorometer after acetone extraction, the fluorescence sensor of an oceanographic buoy, and satellite data. This is interesting because we can compare the results from distinct approaches.
Other innovative content of this manuscript is the place and method of sampling for the in-shore station. One of the authors collected the temperature and seawater samples for Chl a using a surfboard. Although this can be an excuse to surf during working hours, it has a useful part because the wave breaking zones are not investigated received little attention. No one goes to sample with an oceanographic ship in a surf zone due to the wave motion, and oceanographers avoid those environments where expensive instruments can be damaged.
The idea of ​​using surfers to collect oceanographic data is striking, but perhaps more anecdotal than realistic. First, because the area and time preferred by surfers is very limited: areas and periods with big waves. This means a high turbulence, and consequently the values ​​measured at the water surface have a high influence of the atmosphere (air bubbles), in addition to the problem of the precision of instruments placed on a surfboard.
For phytoplankton, with the exception of surf diatoms (i.e., Anaulus), high levels of turbulence, resuspension of mineral particles (sand), and abundant detrital material (i.e., decomposition of macrophytes) is an unfavorable environment for phytoplankton growth. Chlorophyll measurements mostly correspond to non-viable cells. Another problem of this study is that the sample preparation and storage method for chlorophyll analysis is not identical between the offshore and inshore stations. Samples from the inshore station were prepared at home. The size of the filter is different and the conservation method before the analysis at the lab is also different. An important deficiency is that nutrient data are not provided at the inshore station. Sampling for nutrients is not particularly difficult, as well as a Lugol-preserved phytoplankton sample. These data are missing.
There are data from the oceanographic buoy, but according to the authors the buoy was out of the water for winter maintenance (winter is 3 mo). However, the buoy data are missing for almost 6 months, which is a relevant period for a one-year study. The methods of collection and observation of phytoplankton are explained, but the results barely comment on the abundance of a coccolithophorid in a single sampling day. Limitation by light or nutrients is discussed, but there are not experiments, it is only assumed that nutrient depletion implies a limitation. There are several factors that make this study preliminary and with experimental design errors that need to be corrected. The text expands too much on aspects that are not really covered. For example discussing on climate change when only one year of chlorophyll data is provided is unsupported. Paragraphs in the introduction and discussion need to be deleted or shortened because they are not necessary. However, despite these shortcomings, I like graphs comparing different approaches to such important data as phytoplankton biomass (as Chl a), and the multidisciplinary nature and the development of new sampling methods, involving citizens, are welcome in oceanography.
The contents fit with an especial issue on Land-Ocean Interactions.
Specific comments
Title: An unconventional investigation into the seasonal dynamics of phytoplankton in nearshore coastal waters of southwest England"
Please modify the title. There is no data on the seasonal dynamics of phytoplankton, just only values of chlorophyll a. Please try to explain the ‘unconventional investigation’. Half of the title is “in nearshore coastal waters of southwest England”, just reporting “off Plymouth, SW English Channel” is OK. This study is a comparison of distinct phytoplankton chlorophyll a methods in the surf zone and offshore Plymouth, western English Channel.
Abstract
line 5: Nearshore coastal waters often contain the highest levels of biodiversity and phytoplankton biomass.
I agree that we have usually more phytoplankton biomass in coastal waters, but I have doubts that we have more biodiversity of phytoplankton (species richness?). From each 10 cells that you observed in coastal waters, sometimes 9 belong to the same species, while in the open ocean you have more “biodiversity”. If you consider the number of species/number of individuals you have more diversity in the open ocean.
line 6: less is known about the seasonality of phytoplankton in the nearshore compared to offshore coastal, shelf and open ocean waters.
I disagree. There are more numerous phytoplankton studies in the nearshore coastal waters, for example for the monitoring of harmful microalgae, when compared to the open ocean.
line 7: we analyse a unique annual dataset of chlorophyll-a concentration – a measure of phytoplankton biomass
Why unique?
Chlorophyll-a concentration is an estimation, a proxy, more than a direct measurement of the phytoplankton biomass. The chlorophyll a concentration of a cell changes according to the physiological stage and the environmental conditions.
line 11: and guided by satellite observations of light availability
Is it relevant to use a satellite to measure the light? You can use the data of a coastal weather station.
line 16: “Offshore (Station L4) chlorophyll-a concentrations dropped dramatically whereas chlorophyll-a concentrations in the nearshore (Bovsiand Beach) remained high. Statistical comparisons between chlorophyll-a concentration and physical (SST and light) and chemical variables (nutrients) suggest that the offshore site (Station L4) becomes nutrient limited at the surface during the summer, in contrast to the nearshore”.
These are the only results in the abstract. Low levels of Chl a in offshore water during the summer, while the levels are relatively high in inshore waters is a common feature in any temperate sea. Nutrient limitation cannot be only inferred based on a nutrient concentrations.
I am missing in the abstract that the authors will compare the advantages and disadvantages of the distinct proxies of the phytoplankton biomass.
Keywords: Phytoplankton, Phenology, Citizen Science, Coastal, Nearshore
Keywords such as coastal or nearshore are already in the title and abstract. It is more useful to report chlorophyll as keyword because this study reports chlorophyll data.
Introduction
Line 29: Despite constituting less than 1% of the Earth’s photosynthetic biomass [2], phytoplankton are responsible for around half of global primary production.
You are confusing primary producer biomass with photosynthetic biomass. For example, for a palm tree, 90% of the biomass is trunk and roots that is not a biomass able to photosynthesize. Just report that phytoplankton is responsible for around half of global primary production.
line 40: other factors include disease (e.g. viral lysis).
Please provide a reference on the virus.
line 43: whereas at subtropical latitudes, phytoplankton abundance can be higher in winter, though not always in coastal waters
This is unclear, why the phytoplankton abundance can be higher in winter, and the differences between coastal waters and open ocean? Do you have a winter in tropical latitudes?
Please do not enter in these topics. You are measuring the chlorophyll during a year off Plymouth. It is not necessary an introduction to the phytoplankton ecology in the world oceans.
line 60: Satellite based monitoring of ocean colour now provides over two decades of continuous observation, capturing the seasonal cycles of surface phytoplankton.
Very good the use of ‘surface’ phytoplankton because the phytoplankton is below the surface in most of the oceans.
line 65: The coastal zone is regarded as one of the most valuable yet vulnerable habitats on Earth [35]. It has the richest biodiversity of any marine habitat.
When you have a monospecific bloom of dinoflagellates in a coastal zone, you have not a rich biodiversity. I insist, there is more biodiversity in the open ocean.
line 69: monitoring phytoplankton in nearshore coastal regions (e.g. surf zone) is challenging and seldom documented in the literature.
The surf zone is thin line characterized by strong mixing (particles, detritus, low irradiance, air bubbles). That is not the ideal environment for phytoplankton. The surf zone is thin line which role in the global primary production is small. However, any part of the ocean deserves consideration.
line 75: The lack of chl-a data in this region highlights the need to explore and develop new phytoplankton measuring platforms for the nearshore coastal zone.
line 78: For example, it has been estimated that 40 million measurements on environmental indicators per year could be acquired by surfers in the UK [45].
This should be in the title: surfing for collection oceanographic data.
Please report the cost of the equipment in the method.
line 91: At present, there appears to be only a few studies, primarily focused on lakes, that have compared seasonal dynamics in phytoplankton between nearshore and offshore coastal waters.
I do not think that limnologists have paid more attention than oceanographers.
line 119: the presence of a trained scientist (also a surfer) who lived in close proximity to the beach and was willing to collect data.
From now on oceanographers will put on their CV if they are surfers because it is a method to sample.
Line 141: Whatman 47mm 0.7 um GF/F microfibre filters
Please be careful, the filter has not pores of 0.7 µm. You will not see holes of 0.7 µm in the filter. It is estimated that the efficiency is equivalent to a membrane filter of 0.7 µm pore size.
Please note that you are comparing the values of chlorophyll offshore and inshore. However, you are adding variability if you do not use the same filters and filtration equipment, and conservation of the sample.
Line 199: buoy was taken out of the water during the winter for essential maintenance.
The buoy data are missing for almost 6 months in a study of one year. Winter is only 3 months.
line 219: Samples are collected according to the Utermohl counting technique
Wrong, strictly the Utermöhl method is not for sampling, and strictly Utermöhl method is not a counting technique. The Utermöhl method is a combination of a concentration of the phytoplankton cells using a settling column, and a method of observation using an inverted microscope. Utermöhl method means that you used settling columns and an inverted microscope. You must explain how you sampled and how you counted.
line 220: Individual taxa are grouped into Diatoms, Dinoflagellates, Coccolithophores, Flagellates, Phaeocytis and Ciliates.
Where are these data in the text? The only comment is about the abundance of coccolithorids in one day of summer.
Phaeocytis is Phaeocystis. Why Phaeocystis is not a flagellate?
Line 277: Figure 4. The labeling of the x-axis of the graphs is confusing. For example 2017.8 seems to correspond to August 2017, but it is October 2017. Please divide the year into 12 months using the labelling J F M A M J J A S O N D. Please correct this in other figures. For the y-axis, the chlorophyll values varies from 0 to 10. Log10-transformed data are useful for values that vary for several orders of magnitude. It is not needed to use logarithmic values.
line 299: phytoplankton group are known to be strongly associated with stratified waters [93].
Please avoid citing references in the result section.
Figure 8. Surface nutrient concentrations.
You cannot infer nutrient limitation from only the absolute values of nutrient concentrations. A poor proxy is to use the nutrient ratios. The ratio N/Si is high, but not too high (i.e., N:Si ratio of 4:1). Diatoms such as Pseudo-nitzschia have blooms during the summer in other areas of the English Channel. Do you think that the high N/Si ratio in late spring is responsible of the summer bloom of the coccolithophorid Emiliania huxleyi?
329: PAR suggests that it is not light that is limiting phytoplankton growth at L4 during the summer.
Again, to infer light limitation from only the values of irradiance is risky. Nutrient and light limitations are issues that need specific experiments, not only based on the absolute values.
Please do not enlarge the manuscript with light, nutrient limitations or climatic change when you have not data to deal on those important issues. Please focus on the comparison of the distinct methodologies to report the chlorophyll a.
Line 393: For example, under a scenario of enhanced coastal stratification, nearshore phytoplankton may be more resilient than those offshore.
Please explain in more detail. Please note that you do not provide data on the phytoplankton. Are there differences in the phytoplankton composition between the in- and offshore stations?
Author Response
Please find our response to comments from Reviewer 1 in the attached PDF file.

Reviewer 2 Report
This is a rather interesting study addressing the use of citizen science data to help characterized the relatively poorly studied nearshore phytoplankton abundance. The study compares data collected by a platform installed in surf boards (Smartfin) with data collected by a ‘conventional’ platform in adjacent offshore area as well as data provided by remote sensing. The data collected at the two locations included sea surface temperature and solar radiation (PAR), but the most interesting dataset refers to chlorophyll concentration, the common proxy of phytoplankton biomass.
Main results indicate that while nearshore and offshore data coincide for most part of the year, a significant difference could be observed for the summer months. These results are discussed as illustrating the validity and usefulness of citizen science data is enabling a detailed characterization of phytoplankton dynamics in nearshore locations.
The paper is very well written, the methods are sound and carefully tested, and the conclusion draw are supported by the presented data. I can recommend publication and just have one small suggestions of change:
Fig. 6. The unit Einstein is not a SI unit, it was replaced by umol (micro mol) quanta; so the y-axis should display units of umol quanta/m2/s.
Author Response
Please find our response to comments from Reviewer 2 in the attached PDF file.

Round 2
Reviewer 1 Report
The revised version includes some of the suggestions. It is not easy to follow the modifications because authors did not use the track changes function in MS Word.
More than the value of the data (restricted to one year of chlorophyll data), the publication is supported by the originality in the method and place (surf area). Next time, you will need to include more data and to use the same methods for the offshore and near shore stations.
The manuscript is excessively enlarged when compared to the amount of data. For example, the first 3/4 of the abstract is only describing the method. The results are reported in only a single sentence. There are three questions at the end of the introduction, and the abstract should respond to these questions. Instead of a conclusion section, there is a summary at the end of the manuscript. Some contents of the summary should be in the abstract.
It is evident that the manuscript is unnecessarily enlarged because there are 73 cited in the introduction. This manuscript often looks like a review on phytoplankton ecology.
The discussion of nutrient or light limitations, just only based on raw data, without experimental approaches, is unsupported. In addition, there is no nutrient data from the nearshore station. A section for climate change based on only one year of chlorophyll data is unnecessary. The consequence is that extension of the manuscript does not reflect the real amount of new methods and data. This was already commented in the first review, with scarce success. Despite of this, I would recommend the publication because I like the originality sampling in the surf area and the potential interaction between oceanographers and citizens.
Minor comments
title: We are oceanographers, no terrestrial ecologists. England is not a sea, your sea is the English Channel. Please replace Plymouth, England by Plymouth, SW English Channel
line 9: on a near weekly basis between 2017 and 2018 .
It looks a two-year sampling, please report from September 2017 and September 2018
Some people prefer to report Chl-a with a in italic type.
Line 218: all cells >2 mm were identified, and enumerated to the species level (where possible)
Better: all cells >2 mm were enumerated , and identified to the species level (where possible)
You can quantify all the cells because you know that are cells, but you cannot give a scientific name to all the cells, especially the small ones.
line 220: six functional groups; flagellates, diatoms, Phaeocystis, coccolithophorids, dinoflagellates, and ciliates
Taken into account that dinoflagellates contain both heterotrophic and photosynthetic species, dinoflagellates are not a single functional group. In order to avoid the discussion, and remove functional.
Figure 4-7 legends for the x-axis. A year has 12 months, and you have divided the year into 10 parts. It is confusing to report 2018.6 and below 2018-08-07. Just place only the date and avoid confusion.
Line 368 4.3. Implications in a changing climate
It is not necessary this section. Your time series is too short.
The manuscript ended in the section Summary. It is more common to report a section of conclusions. Some of the contents of the summary should go the abstract.
Author Response
Please find our response in the attached PDF.

Reviewer 2 Report
The authors have adequately replied to my suggestions.
Author Response
We are glad to hear. Thanks for reviewing our paper.
Round 3
Reviewer 1 Report
Certainly, it is confusing to have two distinct date formats in the x-axis of the figures 4-7. I have insisted without success.